# Organophosphate Chemical Nerve Agents, Oxidative Stress, and NADPH Oxidase Inhibitors: An Overview

**DOI:** 10.3390/ijms26199313

**Published:** 2025-09-24

**Authors:** Christina Meyer, Thimmasettappa Thippeswamy

**Affiliations:** Department of Biomedical Sciences, College of Veterinary Medicine, Iowa State University, Ames, IA 50011, USA

**Keywords:** Organophosphates, chemical warfare agents, NADPH oxidase, apocynin, diapocynin, mitoapocynin, oxidative stress, reactive oxygen species, epileptogenesis

## Abstract

Organophosphates (OPs) are potent anti-acetylcholinesterase compounds historically used as pesticides and exploited in chemical warfare. Exposure to OPs initiates cholinergic crisis with both peripheral and central effects such as salivation, lacrimation, urination, and defecation (SLUD), and status epilepticus (SE), a prolonged state of seizure. Standard medical countermeasures atropine, oximes, and benzodiazepines reduce mortality, control peripheral symptoms, and terminate SE. However, they do not attenuate the consequences of SE, including neurodegeneration, oxidative stress, neuroinflammation, epilepsy, and associated comorbidities such as cognitive dysfunction. SE induces excessive NADPH oxidase (NOX) synthesis and production of reactive oxygen species, which is a key driver of neurodegeneration and epilepsy. Furthermore, inhibition of NOX in SE-induced epilepsy models reduces neuroinflammation, neurodegeneration, and seizure frequency. Following OP toxicity, treatment with NOX inhibitors diapocynin and mitoapocynin reduced oxidative stress and astrocyte reactivity. This review summarizes the history and development of OPs and the current knowledge on OP toxicity, emphasizing the role of NOX, and the therapeutic potential of NOX inhibitors in treating long-term consequences of acute exposure to OPs.

## 1. Introduction

Organophosphates (OPs) are a class of chemical compounds developed initially as pesticides and later used in chemical warfare, political assassinations, and cases of suicide. They are a group of organic compounds that contain phosphorus and are commonly synthesized through the esterification of phosphoric acid or related derivatives with alcohol [1]. Each year, an estimated 3 million people are exposed to OPs [2]. OP self-poisoning accounts for 19.7% of suicides globally; however, underreporting can make it difficult to pinpoint precise poisoning rates [3]. The majority of OP poisonings occur in developing countries due to less regulatory systems and easier access [4]. Because OPs are fast-acting, highly potent, and challenging to detect, they have been exploited in military operations and terrorist attacks [5,6]. Consequently, OPs have shaped the political landscape since their development for wartime purposes. Treatment options are limited, and real-world outcomes are often fatal [7]. This review aims to provide a historical perspective of OPs and the current understanding of maladaptive mechanisms in OP poisoning and excitotoxicity. Particular focus is on the role of oxidative stress driven by nicotinamide adenine dinucleotide phosphate (NADPH) oxidase (NOX) dysregulation and the efficacy of NOX inhibitors as treatment for OP-induced neuropathology and excitotoxicity.

## 2. Historical Perspectives of Organophosphate Nerve Agents

On 23 December 1936, Dr. Gerhard Schrader made a discovery that altered the course of chemical warfare [8]. Working as a chemist at I.G. Farben in Leverkusen, Germany, Dr. Schrader researched organic fluorophosphorine and OP compounds to develop pesticides [9]. In the process, he was accidentally exposed to a spilled drop of newly synthesized ethyl dimethylamidocyanophosphate and experienced symptoms including constricted pupils, restricted breathing, and headache [10]. By an official Reich ordinance of 1935, I.G. Farben was required to report the event to the German War Office [11]. The chemical, later named tabun, became the first OP in the G-series of chemical warfare agents [8]. Dr. Schrader and his team subsequently developed increasingly toxic nerve agents, sarin (GB, 1938), soman (GD, 1944), and cyclosarin (GF, 1949) [12,13]. Tabun and sarin were heavily produced in Dyhernfurth in 1942 and 1945, respectively, though soman was not manufactured at a large scale [9,11]. Thankfully, OP nerve agents (OPNAs) were not employed during World War II. After the war, the remaining stockpiles of tabun and sarin were either destroyed or retained for research purposes [14].

Dr. Bernard Saunders, based in the UK, synthesized diisopropylfluorophosphate (DFP), first described in 1941 [15]. Less volatile G-series nerve agents, DFP can still cause severe incapacitation or death in humans and remains widely used in laboratory research to model OP exposure in rodents and other animals [16]. The second series of OPs, V-agents, were also developed in the UK in the 1950s. The V-series, including agents VX, VE, VG, VM, and VR (Russian VX) is significantly more lethal as they are roughly ten times more toxic than G-agents [14,17].

The first use of OPNAs in warfare was during the 1980–1988 Iran–Iraq war, when tabun and sarin were used against soldiers and civilians. Over 100,000 people were believed to be exposed to OPs during the conflict [18,19]. In a separate conflict, the Persian Gulf War of 1991, many US returning troops suffered from initially unexplained symptoms, including fatigue and cognitive impairments. Unbeknownst to soldiers, rockets filled with sarin and cyclosarin were destroyed upon the demolition of a bunker in Khamisiyah, Iraq [20]. The aforementioned symptoms, later termed as Gulf War Syndrome, were likely caused by a combination of low-level sarin exposure released from the demolition, pesticides, and oil well fire smoke [21]. Interestingly, per reports, soldiers who took pyridostigmine bromide, an anti-acetylcholinesterase intended as a prophylactic medication, experienced worse outcomes and were at higher risk for Gulf War Syndrome [22]. This effect was possibly due to interactions with low-level sarin exposure and other chemical exposures like the insecticide DEET [23,24].

In addition to the use of OPs in military warfare, OPNAs have been used in acts of terrorism. Terrorists in the 1994 Matsumoto and 1995 Tokyo subway attacks released sarin upon civilians, resulting in more than 1000 cases of OPNA poisoning and 19 deaths [25,26,27]. Around the same period, a new class of nerve agents emerged: Novichok, or A-agents that are even deadlier than V-agents [28]. Examples of the chemical structures of G-agents, V-agents, and A-agents are provided in Figure 1.

It is important to highlight the international efforts for protection against OPNAs. The Geneva Protocol of 1925 prohibited the use of biological and chemical weapons (“asphyxiating, poisonous or other gases, and of all analogous liquids, materials or devices” and “bacteriological methods of warfare”) [33]. However, the protocol did not address the storage and production of such chemical weapons. To rectify these weaknesses, 130 countries signed the Chemical Weapons Convention (CWC) in 1993, prohibiting the development, production, and stockpiling of chemical warfare agents, including OPNAs [34]. The CWC also established a scheduling system of chemical warfare agents outlined in the Annex on Chemicals, citing the G- and V-series as schedule 1, chemicals that have been previously produced as a chemical weapon, pose a high risk to the purpose of the Convention, are highly lethal or toxic, and hold little or no uses for peaceful applications (industry). Novichok was later added to the Schedule 1 Annex on Chemicals in 2019 [35].

## 3. Mechanisms of Organophosphate Poisoning

The following sections provide an overview of the pathological sequelae following OP exposure. Section 3.1 details the immediate acute effects of OP poisoning and currently available medical countermeasures (MCM). Section 3.2 describes the transition from status epilepticus to epileptogenesis and epilepsy. Furthermore, Section 3.3 and Section 3.4 highlight neuronal hyperexcitability and gliosis, respectively, as key drivers of excitotoxicity and ongoing neuropathology. While MCMs can prevent mortality, persistent neurological and behavioral deficits remain, which are integrated through these subsections. This organization mirrors the cascade of pathological events during the initial exposure and epileptogenesis that drive long-term outcomes despite treatment with traditional MCMs. A schematic representing the pathological progression of OP-induced neurotoxicity is provided in Figure 2.

### 3.1. Acute Cholinergic Crisis and Medical Countermeasures

The cholinergic system is highly conserved across species and important in many biological functions, including the respiratory, gastrointestinal, and urinary systems. These effects are mediated by the neurotransmitter acetylcholine (ACh), which acts by binding nicotinic (ionotropic) and muscarinic (G protein-coupled) receptors [36]. Nicotinic (nAChR) and the majority of muscarinic (mAChR; M1, M3, M5) receptors are excitatory [36,37]. Following receptor activation, acetylcholinesterase (AChE) hydrolyzes ACh by forming a transient covalent bond between the serine catalytic residue and the acetyl group, resulting in the breakdown of ACh into choline and acetate [38]. Choline is then recycled into the cell to synthesize new ACh by the enzyme choline acetyltransferase [39]. Inhibition of AChE, such as by OPs, disrupts this cycle, leading to excessive accumulation of ACh at the synapse and continual overstimulation of AChRs, causing cholinergic crisis (illustrated in Figure 3).

Compared to other anti-AChE compounds, OPNAs are especially hazardous because they irreversibly inhibit the enzyme. OPNAs bind and phosphorylate the catalytic serine group of AChE, forming a stable covalent bond that prevents AChE from hydrolyzing ACh [40]. The agent–AChE complex then undergoes an aging process, through dealkylation, which increases the stability of the complex [41]. The binding of OPNAs to AChE leads to a pathological state known as cholinergic crisis [42,43,44]. Several side effects mark cholinergic crisis. Respiratory distress occurs, driven by AChE inhibition in both the central respiratory center nAChRs and peripheral mAChRs [44]. Acute OP-induced mortality is generally attributed to respiratory failure [45,46]. Other peripheral effects include emesis, salivation, lacrimation, urination, defecation, pupillary constriction (miosis), and muscle fasciculations [25,47]. The cardiovascular system is also affected, with bradycardia and hypotension being predominant symptoms [48,49]. In the central nervous system, uncontrollable neuronal firing leads to status epilepticus (SE), or a seizure lasting longer than 5 min [50,51,52].

Traditional medical countermeasures for OP intoxication target the cholinergic system and gamma-aminobutyric acid (GABA)-ergic inhibitory receptors. Atropine sulfate (ATS) is a muscarinic AChR competitive antagonist that mitigates OP effects in the periphery. Still, ATS has a limited ability to cross the blood–brain barrier (BBB) and is ineffective at nicotinic sites [53]. Oximes, another standard countermeasure, dissociate OPs from AChE in non-aged agent–AChE complexes. However, oximes do not cross the BBB, though several groups are examining alternative strategies for oxime delivery, such as nanoparticle technology and ultrasound disruption of the BBB [54,55]. Interestingly, despite limited BBB-permeability, oximes are more effective following VX poisoning than G-agents, since VX forms aged agent–AChE complexes at a slower rate than other OPs [5,56]. The third recommended medical countermeasure is the benzodiazepine midazolam (MDZ), a positive allosteric modulator of GABA receptors. MDZ reduces neurodegeneration, as well as SE severity and behavioral seizures when administered within 10 min of exposure [57]. Accordingly, if promptly administered, ATS, oximes, and MDZ are effective in reducing mortality and even neuropathology to some extent. Critically, however, long-term neuropathological changes persist, including epilepsy, psychological comorbidities, neurodegeneration, and gliosis [58,59,60].

### 3.2. Status Epilepticus-Induced Epileptogenesis

Seizures, a state of synchronized neuronal firing, can be triggered by external factors such as acute brain injury, drug withdrawal or overdose, and exposure to OPNAs and other neurotoxins such as kainic acid (KA) and pilocarpine [61,62,63,64,65]. The key distinction between epileptic and non-epileptic seizures lies in their spontaneity. Following OP intoxication, the initial seizures are non-epileptic as the chemical inhibition of AChE directly drives the hyperexcitation. However, the neuronal damage caused during initial seizures that last longer than 5 min in humans and 20 min in animal models lowers the brain’s baseline threshold for excitation, eventually leading to chronic electrographic abnormalities and spontaneous recurrent seizures [51,66]. For example, a follow-up study of survivors of the Tokyo sarin attacks revealed epileptiform discharges 5 years after exposure despite treatment with MCMs [67]. As per the International League Against Epilepsy, diagnosis of epilepsy is made when a patient experiences two unprovoked seizures more than 24 h apart, or one unprovoked seizure with a ≥60% likelihood of recurrence within the next 10 years [68]. Patients who have been previously exposed to OPs are 3.57 times more likely to develop epilepsy compared to their exposure-free counterparts [69].

The complex process of a non-epileptic brain becoming epileptic is known as epileptogenesis, a process that is triggered by exposure to chemoconvulsants [70]. During this period and beyond, a cascade of changes occurs in the brain, characterized by neurodegeneration, neuroinflammation, and oxidative stress [71,72,73,74,75]. Neuropathology observed in OP-induced acquired epilepsy produces psychological comorbidities. These include anxiety, depression, memory deficits, motor impairments, and post-traumatic stress disorder [76,77,78]. Many researchers propose that successful treatment during the epileptogenesis period could attenuate the long-term consequences of pathological sequelae [79,80,81]. In rats, treatment with the inducible nitric oxide synthase inhibitor 1400 W or Src family kinase inhibitor saracatinib (AZD0530) during epileptogenesis, the first two weeks following exposure, mitigated OP-induced behavioral deficits and cellular dysfunction 3.5 months after SE [82,83]. These findings demonstrate the lasting impact of therapy with disease-modifying agents during epileptogenesis.

### 3.3. Neuronal Hyperexcitation and Neurodegeneration

During SE, GABAA receptors are internalized, whereas glutamatergic receptors increase in surface expression [84]. Furthermore, neuronal death during SE occurs in a time-dependent manner, and inhibitory neurons are vulnerable to functional impairment in their pursuit of intercepting the firing of excitatory neurons [85,86]. This phenomenon is why it is crucial to administer MDZ sooner rather than later after OP exposure. Overactivation of glutamate receptors N-methyl-D-aspartate (NMDAR), α-amino-3-hydroxy-5-methyl-4-isoxazolepropionic acid (AMPAR), and KA during SE contributes to synaptic remodeling in epileptogenesis [87]. As epileptogenesis progresses, excitotoxic changes have structural and functional impacts that can be analyzed using magnetic resonance imaging (MRI) [59,83,88,89]. What is more, quantitative MRI can predict OP-induced neurodegenerative outcomes [90,91].

Seizure-induced glutamatergic overactivation results in the release of damage-associated molecular patterns (DAMPS) that signal glial cells, like microglia and astrocytes, to respond [92]. While neuronal firing propagates seizures, cross-talk between neurons and glial cells sustains chronic excitotoxicity in the brain. In the case of OP poisoning, traditional anti-seizure medications that primarily target neurons are unable to control seizures [93]. Thus, in addition to controlling neuronal excitability, it is clear that oxidative stress and neuroinflammatory processes must also be managed to effectively treat long-term consequences of chemoconvulsant-induced SE.

### 3.4. Gliosis

Glia cells, including microglia and astrocytes, are essential supporting cells in the brain. The role of other glial cells (oligodendrocytes and polydendrocytes) in epileptogenesis is limited and beyond the scope of the current review. Microglia, the resident immune cell of the brain, are first-responders to injury via recognition of DAMPS and pathogen-associated molecular patterns (PAMPS) [94]. DAMPS detected by microglia following seizures include high-mobility group box 1, heat shock proteins, extracellular DNA and ATP, and more [95,96,97,98]. Microglia also sense excess glutamate released during SE [99]. In response, microglia transition into a reactive state, releasing proinflammatory cytokines and chemokines and producing reactive oxygen and nitrogen species (ROS/RNS) [100,101]. Importantly, microglia phagocytose neurons and prune synapses in an aberrant manner, which contributes to excitotoxicity [102]. Following OP poisoning, microglia reactivity is characterized by increased expression of lysosomal marker CD68, ameboid morphology, and cytokine release [82,103,104].

Astrocytes are guardian cells in the brain that regulate neuronal function and synapse formation, contribute to the BBB, and communicate with microglia to respond to stress states [105]. A vital pathway in microglia–astrocyte signaling is through astrocytic release of Complement 3 (C3). C3 acts as an “eat me” signal by marking neurons for synaptic phagocytosis by microglia. In patients with drug-resistant epilepsy and rats exposed to OPs or pilocarpine, C3 was significantly upregulated [106,107,108,109,110]. C3 deficiency in mice is protective against SE-induced memory deficits and neurodegeneration [111,112]. An additional role of astrocytes is to maintain the extracellular environment by regulating glutamate and potassium concentrations. During excitotoxic states, downregulation of glutamate-uptake channels such as EAAT1 and EAAT2, as well as inward rectifying potassium channel 4.1 (Kir 4.1), leads to an accumulation of excitatory extracellular molecules that promote neuronal firing [113,114,115]. A significant reduction in Kir 4.1 has been shown in rats following DFP, though expression of EAAT1 and EAAT2 in astrocytes is unclear [116].

## 4. Oxidative Stress and NADPH Oxidase as a Key Pathological Mechanism

Oxidative stress is a state of imbalance between oxidants and antioxidants in the body wherein the levels of oxidants become dangerously high, leading to an accumulation of ROS and RNS. ROS is a family of highly unstable oxidants, including free radicals such as superoxide, and non-radicals such as hydrogen peroxide [117]. RNS constitutes a group of molecules derived from nitric oxide (•NO) and superoxide, including peroxynitrite, nitrogen dioxide, and nitrite [118]. Elevated ROS/RNS results in DNA alteration, lipid peroxidation, and cell death. Consuming 20% of the body’s oxygen supply, the brain is exceptionally vulnerable to oxidative stress [119]. Moreover, oxidative stress is known to disrupt the excitatory/inhibitory network in the brain, contributing to SE-induced excitotoxicity [120]. For example, oxidative stress leads to dysfunction of parvalbumin-positive inhibitory neurons [121]. Moreover, glial cells become strongly activated and exacerbate excitatory signaling through the release of ROS and disruption of neurotransmitter balance [122,123]. Even with standard MCMs, oxidative stress is observed up to 4–6 months post-OP exposure [58,60]. ROS are generated by NOX, mitochondria, peroxisomes, and other cellular components [124,125,126,127]. As NOX is the major producer of ROS during inflammatory or injury states, it is the central mechanism of interest in this review.

NOX catalyzes the production of ROS by the reduction of O_2_ to hydrogen peroxide (H_2_O_2_) or superoxide anion (O_2_•^−^), which is coupled to the conversion of substrate NADPH to NADP+H [128]. All seven NOX isoforms, NOX1–5 and DUOX1 and 2, are composed of a catalytic core with six or seven transmembrane domains and a cytosolic dehydrogenase domain that binds FAD (flavin adenine dinucleotide) and NADPH [129]. Meanwhile, NOX isoforms hold unique properties. For instance, NOX2 requires translocation of cytosolic subunits p47phox, p67phox, and p40phox to transmembrane subunits p22phox, GP91phox for activation, NOX4 is the only isoform to be constitutively active, and NOX5 and DUOX1/2 activation are calcium-dependent [130,131,132]. Cellular and subcellular localizations of NOX isoforms are distinct and well elucidated in the peripheral nervous system [133]. However, many NOX localizations remain to be confirmed and/or identified in the central nervous system (CNS). An up-to-date understanding of the cellular and subcellular localizations of NOX isoforms in the brain is provided in Table 1.

Knowledge of specific isoforms and their role in brain function and pathology varies. For example, NOX3 is primarily studied in the inner ear, and little is known about its activity in the brain [158]. There is evidence, however, that NOX3 promotes oligodendrocyte differentiation in vitro [145]. Interestingly, NOX5 is absent in rodents, making it increasingly difficult to study. Cortés et al. overcame the limitation by establishing an endothelial NOX5 knock-in mouse model and demonstrated memory deficiency and BBB alteration in aging mice [159]. NOX5 overexpression in human brain microvascular endothelial cells (hCMEC) leads to metabolic alteration and cell apoptosis, further supporting its contribution to BBB function [154]. DUOX1 and 2 are rarely explored in the CNS, though one study demonstrated DUOX upregulation in the Drosophila brain in an Alzheimer’s disease model [160]. Additionally, DUOX has been identified in astrocytes, human neuroblastoma cells, and oligodendrocytes [155,157].

Receiving slightly more attention, NOX1 dysregulation has been associated with neurological conditions including stroke, Parkinson’s disease, and cerebral ischemic injury [161,162,163]. Of all the NOX isoforms, NOX2 and NOX4 especially respond in injury states in the brain [164,165]. In microglia, NOX2 activation leads to cell reactivity marked by increased CD68 expression and amoeboid morphology [116,166]. Along with microglia, NOX2 is persistently expressed in neurons during chronic neuroinflammation, leading to neuronal mitochondria dysfunction, neurodegeneration, and a positive feedback loop of ROS production between the two cell types [167]. In astrocytes, NOX4 promotes ferroptosis via mitochondrial dysfunction [151]. Ferroptosis, or cell death by iron accumulation and lipid peroxidation, has recently been implicated in both epilepsy, both human and animal models, and requires further investigation in OP models [168].

NOX is the chief producer of ROS during SE and epileptiform activity [169]. In the KA model of epilepsy, NOX2 was significantly upregulated in both the cortex and hippocampus 1 and 2 weeks post-SE, respectively. NOX4 expression was also elevated in the cortex and hippocampus at varying timepoints post-SE (cortex, 72 h and 2 weeks; hippocampus 1 and 2 weeks) [146]. In parallel, NOX2 was significantly increased in brain tissue from patients with refractory epilepsy [170]. In the cuprizone remyelination model, NOX4 deficiency in mice increased neuronal excitability ex vivo, and NOX4^−^/^−^ glio-neuronal cultures were more susceptible to magnesium-induced neuronal death. The same group reported pronounced NOX4 expression in human epileptic tissue [152]. Collectively, these findings indicate that NOX2 and NOX4 are key contributors to oxidative stress in epilepsy. Additionally, NOX4 may be neuroprotective in some cases of hyperexcitability as a transient compensatory mechanism, yet contribute to pathology when persistently upregulated. None of the remaining NOX isoforms have been examined for their role in SE or epilepsy.

While oxidative and nitroxidative stress have been identified following OP intoxication, recent consideration has been drawn to the involvement of NOX. Several studies demonstrate the upregulation of GP91phox (NOX2) in DFP-challenged rats [107,110]. However, beyond NOX2, there is little knowledge of other isoforms of NOX or their role in different OPNAs, such as soman. Further investigation into NOX, OPs, and other SE models is warranted to establish novel mechanisms. A schematic of the proposed OP-induced NOX activation and its neurotoxic consequences is provided in Figure 4.

## 5. NOX Inhibitors as Therapeutic Agents in Epilepsy and Organophosphate Poisoning

### 5.1. Apocynin, Diapocynin, and Mitoapocynin

Apocynin (4-hydroxy-3-methoxyacetophenone, APO) is a broad NOX inhibitor derived from the Canadian hemp plant, Apocynum cannabinum. APO has been used in medicine since the 1830s by native American tribes to treat snake bites and dropsies (edema) [171]. APO can also be isolated from the Picrorhiza kurroa plant, native to the Nepalese Himalayas, and has been used in traditional Indian medicine [172]. Given the prevalence of oxidative stress in many disease states, apocynin has been tested in models including cardiovascular diseases, cancer (lung, pancreatic, and others), and diabetes [173]. APO treatment has been explored in a multitude of neurological disorders, including Fragile X Syndrome, cerebral ischemia, and Parkinson’s Disease [174,175,176].

While APO can cross the BBB, its bioavailability in the brain is low. At a dose of 5 mg/kg (i.p.), less than 8% APO was detected in the brain versus blood [177]. Positron-emission tomography (PET) revealed notable brain radioactivity of APO in mice at the high dose of 100 mg/kg (i.p.) but not at lower doses of 10, 25, and 50 mg/kg [178]. Moreover, APO is a phenolic compound, a trait associated with limited bioavailability [179]. To address these concerns, derivatives of APO have been developed to enhance drug efficacy [177,180,181]. Of many synthesized options, DPO and MPO are more commonly investigated in neurological disorders.

APO does not undergo conversion into DPO in vivo [182,183]. Instead, DPO is chemically synthesized by oxidative coupling of two apocynin monomers [184,185]. DPO was once thought to be an active metabolite, though further investigation revealed that DPO is not a metabolite, though it is generally considered more potent than APO [186,187]. DPO is 10 times more potent and selective, and 13 times more lipophilic than APO [188]. DPO was protective against oxidative stress markers in models of muscular dystrophy, Parkinson’s disease, and Huntington’s disease [183,189,190]. Notably, it requires a large dose (300 mg/kg) of DPO to achieve therapeutic efficacy, and it is improbable to extrapolate the dose for human use [107]. To circumvent this problem, mitoapocynin (MPO) was developed, which showed efficacy at a very low dose (3 mg/kg, oral) in the mouse PD model [191]. MPO offers a targeted approach via synthesis by TPP-conjugation to APO to inhibit mitochondrial NOX [192]. In a Parkinson’s disease model, MPO prevented mitochondrial damage and microglial reactivity [193].

### 5.2. APO, DPO, and MPO as a Therapy for Excitotoxicity

Due to their antioxidant and anti-inflammatory properties, APO, DPO, and MPO present promising options for treating SE-induced neuropathology. Several studies have investigated APO in the pilocarpine exposure model, a chemoconvulsant that mimics temporal lobe epilepsy. In rats, APO rescued pilocarpine-induced neurodegeneration and ROS expression in the hippocampus [61]. Replicating and expanding upon these results, Kim et al. reported that APO significantly reduced neurodegeneration, ROS production, oxidative injury (4HNE), and BBB damage in the hippocampus following pilocarpine challenge. These effects were observed at different timepoints (three days, one week, or four weeks) across two separate studies [194,195].

APO has also been tested in the pentylenetetrazol (PTZ) model of generalized epilepsy. APO treatment during or after PTZ kindling mitigated seizure severity, oxidative injury (4HNE), mitochondrial damage, and autophagy [196]. Furthermore, pre-treatment with APO significantly mitigated anxiety and depression-like behaviors as well as memory impairment in PTZ-kindled mice [197]. However, APO treatment effects following SE are uncertain, as a kindling model does not necessarily propagate convulsive seizures longer than 5 min.

Unlike APO, which has not been tested in OP-induced SE models, we assessed the efficacy of DPO and MPO following DFP exposure [107]. In rats, DPO reduced GP91phox, 4HNE, neurodegeneration, astrogliosis, and epileptiform spikes after DFP challenge [107]. MPO reduced reactive oxygen species and inflammatory cytokines in the periphery, and astrogliosis in the rat brain post-DFP [110,116]. Interestingly, MPO was investigated in KA-induced excitotoxic injury, where MPO significantly decreased NOX4 expression and cerebral lesion size. Though KA was given as an intracranial injection to the striatum and did not induce seizures [198], the role of NOX4 inhibition by MPO is unknown in SE-induced brain pathology.

### 5.3. Additional NOX Inhibitors

Very few alternative NOX inhibitors have been evaluated for the treatment of epilepsy. One group discovered a significant reduction in seizure frequency in rats exposed to KA following treatment with gp91ds-tat, a peptide that inhibits NOX2 [199]. Additionally, dextromethorphan, a NOX inhibitor and NDMA antagonist, has demonstrated neuroprotective effects in the KA model [200]. NOX inhibitors GSK2795039 (NOX2-specific) and celastrol reduced KA-induced seizures [201]. Though they were given intracerebroventricularly, further investigation is required to determine if they can bypass the BBB at adequate levels in the KA model for efficacy. Likewise, pretreatment with GSK2795039 reduced seizure severity and hippocampal proinflammatory cytokines in a PTZ seizure model [202]. Setanaxib, a selective NOX1/4 inhibitor, rescued BBB leakage and signs of oxidative stress following seizures in newborn pigs [203]. Notably, alternative NOX inhibitors such as gp91ds-tat, GSK2795039, and setanaxib have not been investigated in OP models, as the impact of NOX in OP poisoning remains a relatively new area of research, and considerations such as BBB penetration and isoform-specific effects have yet to be evaluated.

## 6. Conclusions

OPNAs are toxic compounds that cause long-lasting adverse effects like epilepsy despite treatment with standard MCMs. Oxidative stress, neuronal hyperexcitability, and neuroinflammation by reactive astrocytes and microglia drive the development of spontaneous recurrent seizures and behavioral comorbidities following OP exposure. Notably, oxidative stress by NOX is a key mechanism that underlies both the development and persistence of epilepsy and symptoms. NOX inhibitors such as APO, DPO, and MPO, mitigate excitotoxicity. MPO is a candidate for treating OP intoxication, due to its targeted reduction in reactive astrogliosis in the DFP model. However, further investigation into the role of specific NOX isoforms in OP poisoning is needed to clarify their role in OP-induced pathology.

As NOX is protective in certain circumstances, it would be beneficial to identify if targeting a specific isoform, such as NOX4, would be more effective than broadly inhibiting all isoforms. Furthermore, given that NOX5 is absent in rodents, additional approaches, such as cell-line models of OP poisoning, are required to determine its activity and possible contribution to OP-induced pathology. The compatibility of therapeutic agents in OP poisoning would need to be considered as well. For example, evidence shows BBB alteration following OP exposure [204,205]. Therefore, pharmacokinetic studies specific to OP models are required, including the investigation of potential interactions between standard MCMs and the therapeutic intervention. Additionally, a combination of NOX inhibitors with standard anti-seizure medications or other disease-modifying compounds like saracatinib is a valuable treatment approach to explore. Altogether, there are many questions left unanswered. Nevertheless, given the role of NOX in other neurodegenerative disorders, including epilepsy, it is worth researching NOX-mediated oxidative stress for the treatment of OP neurotoxicity.

## Figures and Tables

**Figure 1 ijms-26-09313-f001:**
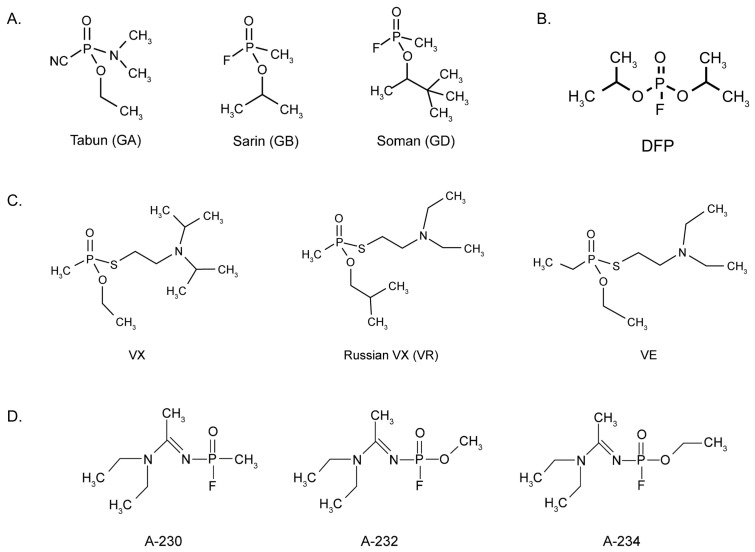
Chemical structures of G-agents (**A**), diisopropylfluorophosphate (DFP, (**B**)) V-agents (**C**), and A-agents (**D**) Within the past ten years, there have been several cases of OPNA poisoning. In 2017, VX was used in the widely known assassination of Kim Jong-Nam [29]. In 2018, Novichok was used in an assassination attempt of a former Russian intelligence officer and his daughter in Salisbury, UK [30]. In a connected event, two civilians in Amesbury, UK, sprayed themselves with a perfume bottle, later determined to contain Novichok [31]. The most recent case was that of a Russian opposition leader exposed to Novichok in 2020 [32]. The intended target victims of both Novichok events survived the attacks due to prompt access to medical care; however, one individual in the Amesbury event succumbed to poisoning.

**Figure 2 ijms-26-09313-f002:**
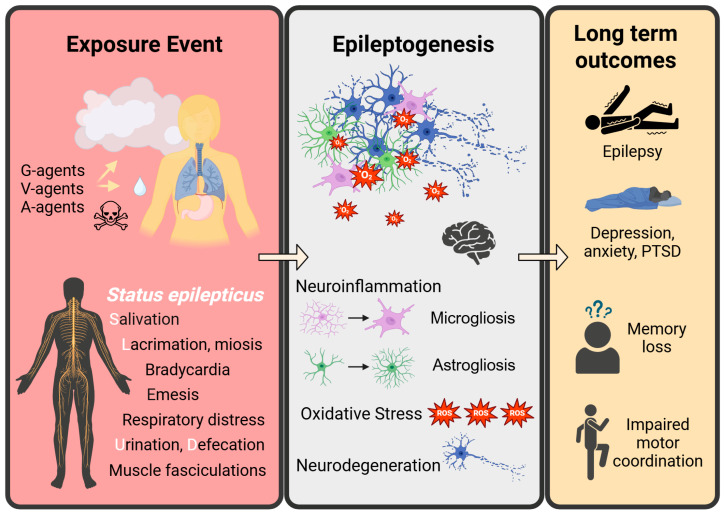
A schematic overview of OP exposure and subsequent pathology. OP intoxication can occur through ingestion, inhalation, or skin contact, leading to cholinergic crisis with symptoms including status epilepticus and SLUD. After the initial event, epileptogenesis, or the development of epilepsy, begins. Epileptogenesis is marked by neuroinflammation, including reactive microgliosis and astrogliosis, oxidative stress, and neurodegeneration, which persists and causes long-term consequences of epilepsy, mental health comorbidities, memory loss (question marks represent impaired recall), and impaired motor coordination.

**Figure 3 ijms-26-09313-f003:**
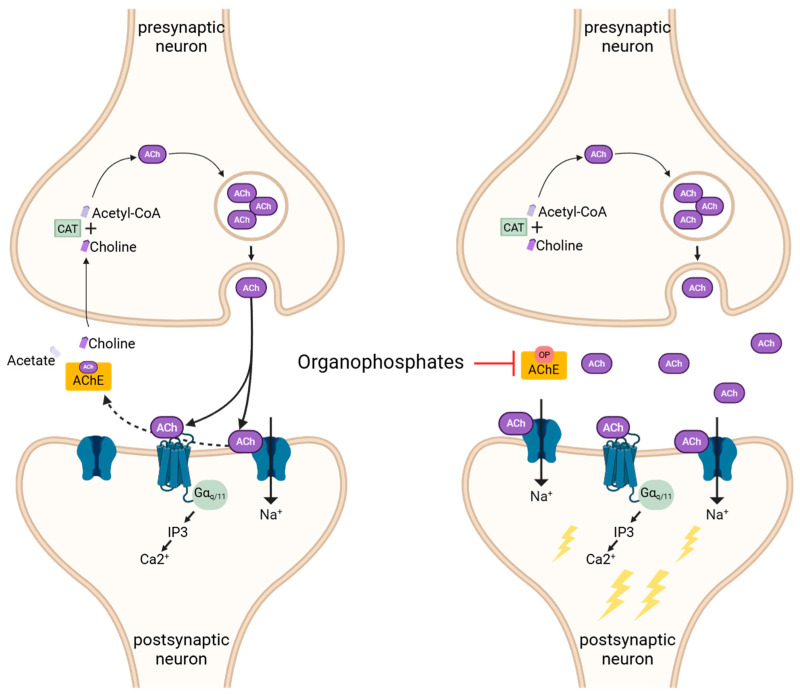
Mechanism of action of OP chemical nerve agents. Acetylcholine (ACh) binds to nicotinic (ionotropic) and muscarinic (G protein-coupled) receptors, creating excitatory post-synaptic potentials. Acetylcholinesterase (AChE) breaks down ACh into choline and acetate, terminating the transmission. Choline is transported into the presynaptic neuron, where choline acetyltransferase (CAT) synthesizes ACh. Organophosphates (OP) form an irreversible covalent bond with the catalytic serine residue of AChE, thereby inhibiting the enzyme, causing an accumulation of ACh, hyperexcitation of the post-synaptic neuron, and cholinergic crisis.

**Figure 4 ijms-26-09313-f004:**
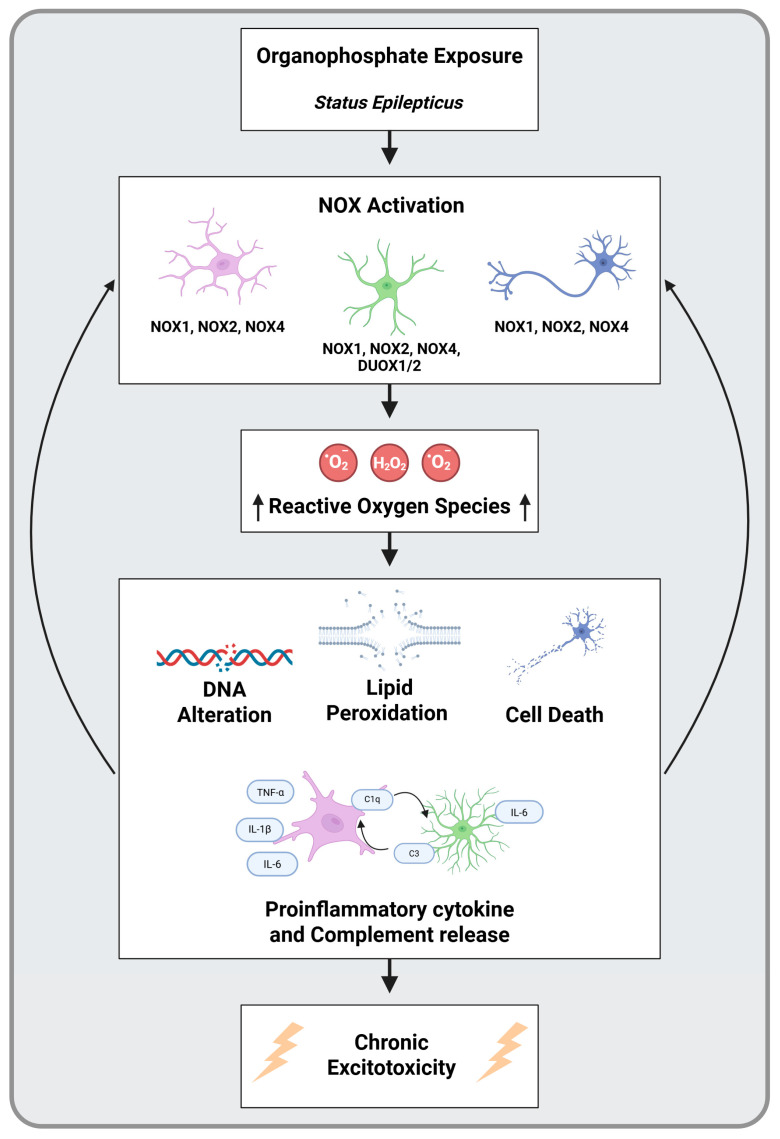
A schematic of organophosphate (OP) exposure, NADPH oxidase (NOX)-mediated oxidative stress. OPs induce NOX activation in microglia (purple), astrocytes (green), and neurons (blue). The reactive oxygen species produced by NOX causes DNA alteration, lipid peroxidation, cell death, and proinflammatory responses leading to a positive feedback loop of ROS production and a chronic state of excitotoxicity. While NOX2 activation is supported by existing evidence, the involvement of other isoforms (e.g., NOX1, NOX4, DUOX1/2) remains uncertain and requires further validation.

**Table 1 ijms-26-09313-t001:** NOX isoforms in the brain.

Isoform	Cell Type	Subcellular Localization	Product
NOX1	Neurons [134], microglia [135], astrocytes [136], neurovascular endothelial [137]	Phagosomal membrane [135], mitochondria [138], nucleus [139]	O_2_•^−^
NOX2	Neurons [140], microglia [110], astrocytes [136], oligodendrocytes [141]	Plasma membrane [142,143], mitochondria-associated endoplasmic reticulum membrane [144]	O_2_•^−^
NOX3	Oligodendrocyte precursor cells [145]	Unknown	O_2_•^−^
NOX4	Neurons [146], microglia [147], astrocytes [148], microvascular endothelial [149], pericytes [150]	Mitochondria [151], mitochondria-endoplasmic reticulum contact sites [152], nucleus [152], cytoplasm [153]	H_2_O_2_
NOX5	Microvascular endothelial [154], oligodendrocyte precursor cells [145]	Unknown	O_2_•^−^
DUOX1	Astrocytes [155], oligodendrocytes [156], neuroblastoma [157]	Unknown	H_2_O_2_
DUOX2	Astrocytes [155], oligodendrocytes [156], neuroblastoma [157]	Unknown	H_2_O_2_

## Data Availability

Not applicable.

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
