# Peer review of "Organophosphate Chemical Nerve Agents, Oxidative Stress, and NADPH Oxidase Inhibitors: An Overview"

_ijms, 2025, doi:10.3390/ijms26199313_

Round 1

Reviewer 1 Report

Comments and Suggestions for Authors

The authors have done a decent effort to synthesize a large body of information, logically progressing from the history of organophosphates (OPs) to the molecular pathology of exposure and the potential of NADPH Oxidase (NOX) inhibitors as a novel therapeutic strategy. The manuscript is timely, well-structured, and highly readable. The figures and table are clear and add significant value. The suggested revisions below are intended to further strengthen the manuscript content.

Major Points for Consideration

Bridging Neuronal Excitotoxicity and Glial Activation: Section 3.3 effectively discusses neuronal hyperexcitation, while Section 3.4 details gliosis. The transition between these sections could be strengthened by explicitly detailing how the initial excitotoxic storm (glutamatergic overactivation) acts as the primary trigger for the subsequent microglial and astrocytic activation. Briefly mentioning the release of DAMPs (e.g., ATP, HMGB1) from dying neurons as the specific signal that initiates the glial response would create a more seamless mechanistic link between these phenomena.

Future Directions and Translational Hurdles: The conclusion is concise and effective. However, the manuscript could benefit from a short, forward-looking paragraph discussing the key challenges and future directions for this line of research. For example:

o             What are the primary hurdles to translating NOX inhibitors into clinical practice for OP poisoning? (e.g., therapeutic window, BBB permeability for newer inhibitors, potential side effects of systemic NOX inhibition).

o             Are there specific NOX isoforms (beyond NOX2 and NOX4) that warrant more immediate investigation in OP models?

o             Could NOX inhibitors be combined with other novel disease-modifying agents (e.g., the Src kinase inhibitors mentioned in the text) for a synergistic effect?

Some Minor Points

  • Section 2 (History): In the discussion of Gulf War Syndrome, the manuscript notes that pyridostigmine bromide was associated with worse outcomes. This is a very interesting point. If space permits, adding a brief clause speculating on the proposed mechanism (e.g., "...possibly due to interactions that enhanced susceptibility to low-level sarin exposure.") would add valuable depth for the reader.
  • Section 4.1 & Table 1: The text mentions that NOX5 is absent in rodents, making it difficult to study. It would be beneficial to explicitly state that this is a significant limitation for preclinical OP research, as results from rodent models may not fully capture the pathology involving this specific isoform.
  • Section 5.3 (Additional NOX Inhibitors): This section does a good job of listing other available inhibitors. It could be slightly improved by briefly commenting on why these agents, particularly the more specific ones like GSK2795039 and Setanaxib, have not yet been evaluated in OP models, and whether they represent promising candidates for future studies.
  • Figure 1: This is a fantastic schematic. The label for "Memory loss" includes three question marks ("???"), which effectively conveys uncertainty. The red coloring is off-beat. Mild/light colors are suggested. The caption could briefly state that memory and cognitive deficits are reported, but the underlying mechanisms are still being fully elucidated, to clarify the reason for the question marks.

Reviewer 2 Report

Comments and Suggestions for Authors

The manuscript by Meyer and Thippeswamy provides a valuable overview of the history and development of OPs, together with the current understanding of OP toxicity. The emphasis on the role of NOX and the potential therapeutic applications of NOX inhibitors in mitigating the long-term consequences of acute OP exposure is of clear interest. The work addresses an important and timely topic; however, I believe that several aspects should be revised to strengthen the manuscript before it can be considered for publication:

  • The inclusion of general chemical structures of OP compounds, along with some representative examples, would greatly enhance the clarity and completeness of the review.
  • Figure 2 requires improvement: the inhibition mechanism of OPs is not sufficiently clear from a molecular perspective. I strongly encourage the authors to describe the relevant chemical reactions, both under normal conditions and in the presence of OPs.
  • The persistence of long-term neurological effects, even after the administration of ATS, oximes, and MDZ, should be further discussed and clarified.
  • The rationale behind the subdivision of Section 3 is not entirely clear; a clearer explanation of the criteria used for this division would be beneficial.

Reviewer 3 Report

Comments and Suggestions for Authors

An excellent review. I have some minor suggestions:

1) Delete (2013, 2018) after Kim et al. in page 9, line 338.

2) Since 4.1. NADPH oxidase (NOX) is the only subheading of Section 4, it could be deleted.

3) I suggest to add a figure indicating the changes of oxidative stress markers in models of exposure to organophosphate compounds
